# Anxiety in Autism Spectrum Disorder: Clinical Characteristics and the Role of the Family

**DOI:** 10.3390/brainsci12121597

**Published:** 2022-11-22

**Authors:** Silvia Guerrera, Maria Pontillo, Maria Cristina Tata, Cristina Di Vincenzo, Domenica Bellantoni, Eleonora Napoli, Giovanni Valeri, Stefano Vicari

**Affiliations:** 1Child and Adolescence Neuropsychiatry Unit, Department of Neuroscience, Bambino Gesù Children’s Hospital, IRCCS, Viale Ferdinando Baldelli, 41, 00146 Rome, Italy; 2Neurorehabilitation Unit, Bambino Gesù Children’s Hospital, IRCCS, 00165 Rome, Italy; 3Department of Life Science and Public Health, Università Cattolica del Sacro Cuore, 00168 Rome, Italy

**Keywords:** Autism Spectrum Disorder, anxiety symptoms, psychiatric comorbidities, parental psychological distress, children, adolescents

## Abstract

Background: Anxiety Disorder (AD) is among the most common psychiatric comorbidity in children and adolescents with Autism Spectrum Disorder (ASD). Likewise, parental psychological distress (PPD) was linked to anxiety symptoms in children and adolescents with ASD. The aim of this study was to characterise, in a sample of children and adolescents with ASD, anxiety symptoms, the functional impairment associated and the presence of PPD. Methods: Participants were divided into three groups based on their diagnosis: children and adolescents with a diagnosis of ASD + AD, others with a diagnosis of AD but without a diagnosis of ASD, and others with a diagnosis of ASD but without a diagnosis of AD. Results: Group ASD + AD showed lower global functioning than Group ASD and Group AD. Generalised Anxiety Disorder, Separation Anxiety Disorder and Specific Phobias were more frequent in Group ASD + AD. Our findings also showed higher depressive symptoms in Group ASD + AD, both in the child and parent reports. Finally, parents of the Group ASD + AD revealed higher levels of PPD. Conclusions: Our findings suggest that early assessment of AD with functional impairment associated with the role of PPD could define individualised treatments and consequently mean a better prognosis in children and adolescents with ASD and AD.

## 1. Introduction

Autism Spectrum Disorder (ASD) is a neurodevelopmental disorder characterised by persistent deficits in social communication and social interaction, as restrictive and repetitive patterns of behavior, interests, or activities that cause clinically significant impairment in several areas of functioning [1]. ASD is a severe cause of morbidity deriving from early-onset, lifelong persistence, high level of associated impairments, and absence of effective treatment for communication, social and cognitive deficits. The clinical expression of ASD varies wildly, depending on the severity of autistic symptoms and the developmental level. In 2018, about 1 in 44 children (age: 8 years old) received a diagnosis of ASD, according to the Centers for Disease Control and Prevention’s (CDC) Autism and Developmental Disabilities Monitoring (ADDM) Network [2].

In children and adolescents with ASD, psychiatric comorbidity rates are approximately 70–75% [3,4,5], or even 83% [6,7]. Recently, Hossain and colleagues [8] pointed out the high variability of prevalence found in different studies (from 54.8% up to 94%). Psychiatric comorbidities increase the possibility of worse long-term outcomes and impaired quality of life [3,6]. Some studies showed that Anxiety Disorder (AD), Oppositional Defiant Disorder (ODD), and Attention Deficit Hyperactivity Disorder (ADHD) are among the most common psychiatric comorbidities in children and adolescents with ASD. For example, Simonoff et al. [4] examined 112 ASD children (10–14 years) and proposed that at least a third of the participants reported three or more psychiatric disorders in comorbidity. Specifically, 29.2% of the total sample reported AD with symptoms of social anxiety.

Brookman-Frazee et al. [7] investigated psychiatric comorbidities in a group of 201 children with ASD (9–13 years). The most frequent pattern of comorbidity was AD with ADHD and ODD (17%), followed by AD with Mood + ADHD + ODD (16%). Lecavalier et al. [9], in a sample of 658 children with ASD (mean age: 7.2 years), observed that 42% of the total sample reported AD, 81% ADHD, 46% ODD, and 12% Conduct Disorder in comorbidity. In addition, 46% of children with ASD + ADHD met the criteria for anxiety disorders. Taken together, these studies have focused mainly on the prevalence of psychiatric comorbidities in children and adolescents with ASD neglecting their clinical characteristics and significance. Concerning anxiety disorders in ASD, to identify specific treatments, it is crucial to understand which anxiety disorders are most frequently observed and clarify their peculiar clinical characteristics in children and adolescents with ASD. Additionally, it is important to explore the presence (or not) of impairment in global functioning (e.g., social, scholastic, and family contexts) associated with anxiety disorders. Additionally, the role of Parental Psychological Distress (PPD) in families of children and adolescents with ASD and AD should not be underestimated. PPD is defined as family members’ distress, resulting in high levels of family conflict and increased expressed emotions, defined as criticism, hostility, and emotional over-involvement [10,11]. Parents of children with ASD have higher parental stress levels and lower quality of life than parents of normotypical children [12,13,14,15], similar to parents with disabled children (e.g., Down syndrome, cerebral palsy, and intellectual disability) [16,17,18]. Yorke et al. [19], in a systematic review, examined the relationships between the children’s emotional and behavioral problems and PPD in a group of ASD children, finding a moderate association between parenting stress, parental mental health problems, and the ASD child’s emotional and behavioral problems. Taken together, these studies have focused mainly on the prevalence of psychiatric comorbidities in children and adolescents with ASD neglecting their clinical characteristics and significance and the role of PPD. Concerning anxiety disorders in ASD, to identify specific treatments, it is crucial to understand which anxiety disorders are most frequently observed and clarify their peculiar clinical characteristics in children and adolescents with ASD. Additionally, it is important to explore the presence (or not) of impairment in global functioning (e.g., social, scholastic, and family contexts) associated with anxiety disorder and the role of the parental psychological profile. Based on the literature described, the main aim of this study was to deepen the clinical profile of children and adolescents with the co-occurrence of ASD + AD focusing on the characteristics of anxiety disorders in this clinical population, the functional impairment associated, and the possible presence (or not) of high levels of PPD. We propose to test the hypothesis that anxiety disorders in ASD can be considered a separate and distinct construct from the main symptoms of ASD and therefore worthy of clinical attention through interventions aimed at children or adolescents but also at their parents.

## 2. Material and Methods

### 2.1. Participants

Seventy-five children and adolescents (mean age: 11.8 years, standard deviation (SD): 2.3 years; range age: 8–16 years; males: 64; females: 11) and their parents were recruited between May 2020 and March 2021 at the Child Neuropsychiatric Unit of the Bambino Gesù Children’s Hospital, Rome, Italy. All children and adolescents were outpatients attending our Unit for Clinical Assessments. We included participants with only ASD diagnosis, participants with only AD diagnosis and participants with diagnosis of both. Our diagnosis follows the DSM-5 criteria [1]. For all participants, the inclusion criterion was an Intellectual Quotient (IQ) higher than 70.

### 2.2. Procedures

Trained child neuropsychiatrists, psychologists and neuropsychologists conducted neuropsychological and psychopathological evaluations according to the international best practice guidelines for the assessment of neurodevelopmental disorders.

#### 2.2.1. Children and Adolescents Assessment

All participants (N = 75) were assessed with Social Communication Questionnaire (SCQ) [20], a screening instrument that helps evaluate communication skills and social functioning. SCQ is a caregiver report, derived from Autism Diagnostic Interview-Revised (ADI-R) [21] used to assess social communication impairment, the presence of repetitive and restrictive behaviors and screen ASD symptoms. The cut-off recommended by the SCQ manual [20] was ≥15. Of the 75 participants, 44 scored at or above this cut-off. For these, the presence of ASD was evaluated by the Autism Diagnostic Observation Schedule-Second Edition (ADOS-2) [22]. It is a semi-structured assessment tool considered a “gold standard” for collecting standardised information about social communication skills, restricted interests and repetitive behaviors. ADOS-2 was administered and scored by licensed clinicians who have reached clinical reliability on the instrument. The calibrated severity score of each domain was also calculated [23,24]. In order to reach a diagnosis of ASD on the basis of criteria used in clinical practice (both directly administered evaluation and parent report) we chose to include both SCQ and ADOS assessments. The presence of anxiety disorders and other psychopathological disorders was assessed by the Schedule for Affective Disorders and Schizophrenia for School-Aged Children Present and Lifetime Version DSM-5 (K-SADS-PL DSM-5) [25], a semi-structured interview based on DSM-5 criteria [1]. Functional impairment due to neuropsychiatric disorder was assessed by the Children’s Global Assessment Scale (CGAS) [26], a scale based on a score from 0 to 100 (from severe impairment to superior functioning) ranging from 1 (constant supervision) to 100 (functioning above the norm in all areas). Furthermore, all participants and caregivers completed self and parent reports to assess anxiety and depressive symptoms through the Multidimensional Anxiety Scale for Children-Second Edition (MASC–2) [27] and the Children’s Depression Inventory-2 (CDI–2) [28], respectively. Finally, based on the patient’s collaboration and language development, cognitive functioning (IQ) was assessed by Wechsler Intelligence Scale for Children Fourth Edition (WISC-IV) [29] the Leiter-3 [30] or the Raven Matrix [31]. Specifically, in our sample, two participants were evaluated with the Leiter-3 [30], five with Raven Matrix [31] and sixty-eight with the WISC-IV [29]. The WISC-IV provides a measure of global intelligence quotient (IQ), obtained through four different indexes: Verbal Comprehension Index (VCI), Perceptual Reasoning Index (PRI), Working Memory Index (WMI), and Processing Speed Index (PSI). In case of failures in completing the WISC-IV for the inadequacy of the language, we administered Leiter-3 [30] or Raven Matrix [31]. Leiter-3 [30] provides a measure of nonverbal intelligence based on four subtests: Figure Ground, Form Completion, Classification and Analogies, and Sequential Order. Raven Matrix [31] provides a measure of non-verbal intelligence, through the completion of a matrix reasoning test. For all the measures used, the versions of the test validated in Italian were proposed to all the participants. Details on socio-demographic characteristics and children and adolescent assessment data are shown in Table 1.

#### 2.2.2. Parents’ Psychopathological Distress Clinical Assessment

Each caregiver completed the Parenting Stress Index-Short Form (PSI-SF) [32] a self-report questionnaire to investigate parenting distress perceived, examining personal factors, parent–child interaction, and behavioral characteristics of the child. Moreover, caregivers completed the Symptom Checklist 90-Revised (SCL-90-R) [33], a self-report checklist that examined the internalisation and externalisation of symptoms. SCL-90-R was composed of nine principal symptomatologic dimensions: Somatisation (disease linked to bodily disrespect); Obsessive-Compulsive symptoms; Interpersonal Sensitivity (feelings of inadequacy and inferiority); Depression; Anxiety; Hostility; Phobic Anxiety; Paranoid Ideation; Psychoticism (interpersonal alienation). The SCL-90-R is additionally composed of three global indexes: Global Severity Index (GSI), a summary index based on the number of reported symptoms and the intensity of experienced discomfort; Positive Symptom Distress Index (PSDI), which examined the accentuation or minimisation of responses; Positive Symptom Total (PST), a measure of the number of reported symptoms.For all the measures used, the versions of the test validated in Italian were proposed to all the participants.

### 2.3. Statistical Analysis

Data were analysed using SPSS IBM Statistics version 20 statistical software (IBM Corp, Armonk, NY, USA). Three groups’ comparisons based on one-way ANOVA were performed on demographic and psychiatric variables to confirm that groups are comparable for age, IQ and autistic symptomatology, and analysed differences between groups in clinical variables (self-report and parent-report). Post-hoc analyses were performed to determine Bonferroni confidence intervals (95%) to establish differences between means. Chi-square tests were performed on frequency data. The two groups were unequal in size, but Levene’s test confirmed the homogeneity of variance and the Shapiro–Wilk test confirmed the normal distribution of the variables based on continuous data.

## 3. Results

### 3.1. Sample Characteristics

The total sample of 75 children and adolescents (mean age: 11.8 years, standard deviation (SD): 2.3 years; range age: 8–16 years; males: 64; females: 11) was divided into three groups:-Group 1 (ASD + AD) was composed of 21 participants diagnosed with Autism Spectrum Disorder with Anxiety Disorders in comorbidity (18 males, three females; mean age: 11.9, 2.4 years);-Group 2 (AD) was composed of 31 participants diagnosed with only Anxiety Disorders (23 males, eight females; mean age: 11.9, 2.2 years);-Group 3 (ASD) was composed of 23 participants diagnosed with only Autism Spectrum Disorder (23 males, 0 females; mean age: 11.6, 2.3 years).

The three groups did not differ significantly in terms of chronological age (F (2,72) = 0.41; *p* = 0.6637) and IQ (F (2,72) = 2.06; *p* = 0.1353). 

SCQ results confirmed significant differences between the three groups (F (2,72) = 14.92; *p* = 0.000). In particular, Group 2 reported lower scores compared to Group 1 and Group 3 (Gr1 vs. Gr 2: *p* = 0.000; Gr2 vs. Gr3: *p* = 0.000). Other comparisons were not significant (Gr1 vs. Gr3: *p* = 1.000). 

Group 1 and Group 3 were assessed also with ADOS-2. No significant differences were found between the two groups for ADOS-2 total score (F (2,72) = 0.78; *p* = 0.3822), ADOS-2 Social Affect Score (F (2,72) = 0.03; *p* = 0.8569), and ADOS-2 Restricted and Repetitive Behavior score (F (2,72) = 0.05; *p* = 0.8254).

Specific means and SD data are reported in Table 1, separate for each group.

### 3.2. Comparisons between Three Groups (ASD + AD, AD, ASD)

Regarding psychopathological variables, we found significant group differences in functional impairment (F (2,72) = 49.204; *p* = 0.000) according to CGAS. Specifically, Group 1 reported higher functional impairment compared to Group 2 and Group 3 (Gr1 vs. Gr 2: *p* = 0.000; Gr1 vs. Gr3: *p* = 0.000; Gr2 vs. Gr3: *p* = 0.000). Regarding the MASC–2 self-report total scores, no significant differences between the groups were found (F (2,72) = 1.425; *p* = 0.247). Instead, significant differences between the groups were found in the MASC–2 parent report total score (F (2,72) = 10.609; *p* = 0.000), with Group 1 reporting a higher total score compared to Group 2 and Group 3 (Gr1 vs. Gr 2: *p* = 0.015; Gr1 vs. Gr3: *p* = 0.000). Other comparisons were not significant (Gr2 vs. Gr3: *p* = 0.128). Regarding the CDI–2 self-report total scores, significant differences were found between the three groups (F (2,72) = 103.745; *p* = 0.000), with Group 1 reporting a higher total score compared to Group 2 and Group 3 (Gr1 vs. Gr 2: *p* = 0.000; Gr1 vs. Gr3: *p* = 0.000; Gr2 vs. Gr3: *p* = 0.000). Additionally, in the CDI–2 parent report total scores, significant differences between groups (F (2,72) = 8.624; *p* = 0.000) were found, with Group 1 reporting a higher total score compared to Group 2 and Group 3 (Gr1 vs. Gr 2: *p* = 0.003; Gr1 vs. Gr3: *p* = 0.001). Other comparisons were not significant (Gr2 vs. Gr3: *p* = 1.000).

#### 3.2.1. Comparison between Group 1 and Group 2 in Anxiety Clinical Profile

According to K-SADS-PL DSM-5, there were significant differences between Group 1 and Group 2 in the percentage presence of Anxiety Disorders (χ^2^ = 23.4236; *p* = 0.00003). Even if both groups reported elevated percentage frequencies of Generalised Anxiety Disorder (Gr 1: 86%; Gr2: 87%), Group 1 reported higher percentage frequencies of diagnosis of Separation Anxiety Disorder (Gr 1: 38%; Gr2: 10%) and Specific Phobias (Gr 1: 38%; Gr2: 10%), while Group 2 reported higher percentage frequencies in the diagnosis of Social Anxiety Disorder (Gr 1: 9%; Gr2: 10%). No participant received a diagnosis of Selective Mutism, Panic Disorder or Agoraphobia. We summarised these results in Table 2. 

For details, intragroup analyses were conducted to analyse if, within confronted groups (Group 1: ASD + AD and Group 2: AD), there were significant differences in the frequency distribution of different Anxiety Disorders based on chronological age. Considering Group 1 (ASD + AD), significant differences were found when comparing by age (χ^2^ = 23.0153; *p* = 0.00004). In particular, the ≤11 years participants reported a higher perceptual frequency of Separation Anxiety Disorder compared with the >11 years participants. On the contrary, Specific Phobias were majorly reported in the >11 years participants compared to the ≤11 years participants. There were no significant differences between Generalised Anxiety Disorder and Social Anxiety Disorder. Additionally, in Group 2 (AD), significant differences were found when comparing age (χ^2^ = 34.1926; *p* = 0.00001). In particular, in this group, a higher perceptual frequency of Separation Anxiety Disorder was found in the ≤11 years participants compared to the >11 years participants. In contrast, the >11 years participants reported higher perceptual frequency of Social Anxiety Disorder and Specific Phobias compared to the ≤11 years participants. No significant differences were found for Generalised Anxiety Disorder.

We summarised these results in Table 3.

#### 3.2.2. Parental Psychological Distress

Regarding parent assessment, when analysing the PSI-SF results, significant differences between groups were found in the mothers’ results (F (2,72) = 5.455; *p* = 0.006). In particular, Group 1 mothers reported higher scores compared to Group 2 mothers (Gr1 vs. Gr 2: *p* = 0.005), while other comparisons were not significant (Gr1 vs. Gr 3: *p* = 0.145; Gr2 vs. Gr3: *p* = 0.723). Father results in the PSI-FS total score were not significant (F (2,72) = 2.371; *p* = 0.101). Regarding the SCL-90 GSI score, mothers reported significant group differences between groups (F (2,72) = 6.655; *p* = 0.002). In particular, Group 1 mothers reported higher scores compared to Group 2 mothers (Gr1 vs. Gr 2: *p* = 0.002), while other comparisons were not significant (Gr1 vs. Gr 3: *p* = 0.492; Gr2 vs. Gr3: *p* = 0.109). Additionally, in fathers’ SCL-90 GSI scores, significant differences between groups (F (2,72) = 7.999; *p* = 0.001) were found. In particular, Group 1 reported a higher total score compared to Group 2 and Group 3 (Gr1 vs. Gr 2: *p* = 0.001; Gr1 vs. Gr3: *p* = 0.008). Other comparisons were not significant (Gr2 vs. Gr3: *p* = 17.389). Specific means and SD data were reported in Table 2, separate for each group. The results of the comparisons between the three groups in parent assessment scores are shown in Table 4.

## 4. Discussion

The first aim of our study was to deepen the clinical profile of children and adolescents with ASD and anxiety disorders focusing on the characteristics of anxiety in this clinical population, the functional impairment associated, and the possible presence (or not) of high levels of PPD. Based on this goal, we compared three groups of children and adolescents (Group 1: ASD + AD, Group 2: AD, Group 3: ASD) matched by age, IQ and severity of symptoms of ASD. In line with current evidence that recommends the use of multiple assessment modalities and informants to assess anxiety in children with ASD [34], we proceeded to include data from clinical interviews and rating scales gathered from multiple informants (e.g., child and parent) to evaluate how much anxiety symptoms interfere with daily functioning and if this impairment involves one or more contexts. The first result we obtained was the presence of a significant global functional impairment in ASD + AD, compared to the AD and ASD groups. This result is in line with the literature that shows that CGAS scores decreased significantly, secondarily to the increment of the number of psychiatric comorbidities [35]. Additionally, it supports the idea that anxiety disorders in ASD can be considered a separate and distinct construct from core symptoms of ASD that negatively affects children’s global functioning. In addition, our results showed significant differences in the MASC–2 parent report total score, with the parents of Group ASD + AD reporting a higher level of anxiety in these children and adolescents compared to the parents of Group AD and the parents of Group ASD. This is firstly consistent with the idea of co-occurrence of ASD-AD, in which anxiety disorder is a distinct construct. However, we found no difference between the three groups in the MASC–2 self-report, with Group ASD + AD reporting levels of anxiety symptoms similar to the other groups. The literature has already reported a significant discrepancy between child and parents’ ratings on the anxiety measures for children with ASD [36]. We propose that the presence of a primary diagnosis of autism may have reduced insight into the emotional difficulties (e.g., anxiety symptoms) of these children and adolescents, in line with the results of other studies [37,38]. Concomitantly, high vulnerability towards depression among children and adolescents with ASD, frequently with anxiety symptoms [39], generated our interest in exploring depressive symptoms. The use of parent and child versions of CDI-2 is widely applied to investigate depressive symptoms in children and adolescents with ASD. Interestingly, our findings showed higher depressive symptoms in Group ASD + AD than in the AD and ASD groups, both in the child and parents’ reports. These results are in line with existing literature [6,40,41] that showed that rates of depressive symptoms are high among individuals with ASD. Indeed, because of the high level of anxiety symptoms and the consequent avoidance performed due to it, children and adolescents with ASD and AD cannot perceive themselves as being able to cope with their anxiety symptoms and the associated threats; this could lead to a tendency to experience depressive symptoms with a negative self-image. However, this explanation should be taken with caution and further studies are needed to support it. 

Moreover, in our study, we tried to define the clinical characteristics of anxiety in children and adolescents with ASD. Following a systematic process of assessment proposed by [42], we evaluated the symptoms of anxiety in ASD using a clinician behavioral observation (K-SADS-PL DSM-5), in addition to the child and parent report information. We thus obtained an objective evaluation of the clinical symptoms of anxiety, considering the diagnostic categories of the DSM-5 [43]. Our results showed higher symptoms of Generalised Anxiety Disorder in both groups, ASD + AD, and AD. Additionally, we found a higher prevalence of symptoms of Separation Anxiety and Specific Phobias in Group ASD + AD. Whereas, for Group AD, prevalent social anxiety symptoms emerged. These results are in line with the most common comorbid anxiety disorders found in the ASD population, including Specific Phobias, Generalised Anxiety Disorder, and Separation Anxiety Disorder [43]. Gjevik E. and colleagues [44] revealed the prevalence of Specific Phobias, as reported by other authors [45,46,47], but differently from our results, no symptoms of Generalised and Separation Anxiety Disorders were evidenced. In opposition, Esther Ben-Itzchak and colleagues [48], in a group of adolescents with ASD, reported a significant prevalence of Separation Anxiety followed by social and generalised anxiety. It is possible that rate differences could reflect sample variability and different ways of evaluating symptoms of anxiety [43]. 

In addition, intra-group analysis evidenced high Separation Anxiety symptoms in children under 11 years in both groups, ASD + AD and AD, and high Specific Phobia symptoms in children aged 11 years and over for the ASD + AD group. Prevalent social anxiety was found only in Group AD in older children. No significant differences were found for Generalised Anxiety Disorder in both Group ASD + AD and AD. Research examining the influence of the child’s age on anxiety symptoms in ASD has produced mixed results. In a cross-sectional study of toddlers, children, young adults, and older adults with ASD, Davis, Hess, et al. [49] suggested that the trajectory of anxiety symptoms in ASD is similar to that of young people with typical development. Our results confirm the presence of Separation Anxiety in younger children of both groups, rather than a subsequent prevalence in older children of Specific Phobia and social anxiety in the ASD + AD and AD groups, respectively. In opposition, other studies reported no age differences in individuals with ASD [50,51,52,53,54]. The characteristics of the group studied and the assessment methods used may be important conditions to consider to define anxiety symptoms in ASD [55]. Concerning symptoms of generalised anxiety, it is not clear why younger groups may exhibit similar symptoms to their older age counterparts. It is possible that children with ASD, similar to children with TD, are more vulnerable to worry in general and maybe more easily submitted to environmental factors or stimuli anxiety-provoking at that young age, as suggested by Varela et al. [56]. Finally, our study investigated the role of Parental Psychological Distress (PPD) in families of children and adolescents with ASD and AD. Our results showed that the mothers of Group ASD + AD reported higher stress levels, as confirmed by higher PSI-SF scores compared to the AD group. This result is in line with other studies that evidence the presence of reduced parental acceptance and greater use of psychological control among parents of children with ASD compared to parents of TD children [57,58]. Moreover, mothers and fathers of the ASD + AD group showed significantly high SCL-90 GSI scores. In particular, mothers of the ASD + AD group reported higher scores than the AD group. Fathers of the ASD + AD group demonstrated higher scores than the AD and ASD groups. Similar to this result, a longitudinal study demonstrated that mothers of ASD adolescents experienced higher levels of depression, anger, and anxiety compared to mothers of TD adolescents [59].

### 4.1. Clinical Implication and Future Directions

According to the results and in line with current evidence, we believe that mental health practitioners should evaluate the complex psychopathological relationship between ASD and other co-occurring disorders with the impairment of quality of life in terms of functioning in social, scholastic, and family contexts of children and adolescents with ASD as well as their families. Future studies should focus on researching comorbid psychiatric features present in ASD children and adolescents, investigating general psychopathology and exploring possible forms of PPD.

### 4.2. Strengths and Limitations

One of the strengths of this study is that it examined anxiety symptoms in children and adolescents with ASD in detail, identifying the presence of specific clinical features characterising a comorbid phenotype ASD + AD, using a ‘gold standard’ instrument for the assessment of psychiatric disorders (K-SADS-PL DSM-5). Moreover, we examined the level of global functioning using tools based on the clinician’s judgment (C-GAS). Moreover, to our knowledge, this is the first study that investigates the presence (or not) of high PPD in children and adolescents with ASD and AD. This study also has several limitations. Firstly, the sample studied comprises a small number of participants. Therefore, we believe that expanding the study population is necessary to confirm the present data, which can be interpreted as preliminary results. Secondly, the study participants recruited have average IQs. Considering that at least 50% of the ASD population has an IQ of less than 70, we are aware that this choice eliminates a substantial group of individuals with ASD. On the other hand, this choice allowed us to exclude clinical characteristics of anxiety related to other comorbidities, such as intellectual disability. Therefore, specific studies on IQ-impaired samples could be carried out in the future. Moreover, IQ was assessed by three different scales based on the patient’s collaboration and language development. Studies included homogeneous groups and therefore the use of a single scale of cognitive functioning is desirable. Finally, no data were collected regarding parents’ demographic features, such as gender, age and socio-economic background. Therefore, it may be appropriate to expand these parent clinical features, such as knowing much more about the impact of anxiety symptoms on their PPD. In summary, we hope that the goal of future studies will be the deeper investigation of the role of the psychological profile of parents on psychiatric comorbidities in children and adolescents with ASD.

## 5. Conclusions

According to the current literature evidence and the results of our study, we believe that the assessment of ASD in children and adolescents should focus on the early recognition of ASD symptoms, the investigation of psychiatric comorbidities and global functional impairment, in addition to searching for parental psychopathological distress. In this way, it could be possible to plan a tailor-made intervention based on the child’s characteristics that involves parents and caregivers.

## Figures and Tables

**Table 1 brainsci-12-01597-t001:** Socio-demographic characteristics and children and adolescent assessment data scores separately for three groups.

Variable	Group 1	Group 2	Group 3	*p*-Value
	ASD + AD	AD	ASD	
	N = 21	N = 31	N = 23	
	Mean (SD)	Mean (SD)	Mean (SD)	
Age (years)	11.9 (2.4)	11.9 (2.2)	11.6 (2.3)	0.6637
IQ level	96.7 (15.0)	104.6 (12.5)	98.7 (17.4)	0.1353
SCQ total	12.9 (6.6)	4.2 (2.8)	12.1 (9.3)	0.0000 *
ADOS 2 total	5.1 (1.5)		5.5 (1.7)	0.3822
ADOS 2: social affect	5.4 (1.6)		5.5 (1.9)	0.8569
ADOS 2: restricted/repetitive behaviors	5.7 (2.1)		5.8 (2.6)	0.8254
C-GAS	41.7 (3.6)	56.6 (4.0)	48.7 (7.7)	0.0000 *
MASC 2-self total	60.6 (13.7)	58.1 (12.7)	54.5 (9.6)	0.247
MASC 2-parent total	78.7 (17.6)	66.2 (14.7)	57.2 (14.6)	0.0000 *
CDI 2-self total	72.9 (5.3)	52.8 (7.9)	45.2 (5.4)	0.0000 *
CDI 2-parent total	67.8 (13.4)	57.1 (11.1)	54.7 (8.8)	0.0000 *

IQ: intelligence quotient; SCQ: Social Communication Questionnaire; ADOS 2: Autism Diagnostic Observation Schedule-Second Edition; C-GAS: Children’s Global Assessment Scale; MASC 2: Multidimensional Anxiety Scale for Children-Second Edition, self-report and parent version; CDI 2: Children’s Depression Inventory-Second Edition, self-report and parent version. * *p* < 0.05.

**Table 2 brainsci-12-01597-t002:** Frequencies and percentage frequencies of anxiety disorders (K-SADS PL DSM-5) calculated separately for Group 1 (ASD + AD) and Group 2 (AD).

	Gr 1	Gr 2
	N = 21	N = 31
K-Separation Anxiety Disorder	8 (30%)	3 (10%)
K-Social Anxiety Disorder	2 (9%)	3 (10%)
K-Specific Phobia	8 (38%)	3 (10%)
K-Generalised Anxiety Disorder	18 (86%)	27 (87%)

**Table 3 brainsci-12-01597-t003:** Frequencies and percentage frequencies of anxiety disorders (K-SADS PL DSM-5) calculated for Group 1 and Groups based on age differences.

Gr 1		
	≤11 Years (N = 11)	>11 Years (N = 10)
Separation Anxiety Disorder	6 (54%)	2 (20%)
Social Anxiety Disorder	1 (9%)	1 (10%)
Specific Phobias	3 (27%)	5 (50%)
Generalised Anxiety Disorder	9 (81%)	9 (90%)
**Gr 2**		
	**≤11 Years** **(N = 16)**	**>11 Years** **(N = 15)**
Separation Anxiety Disorder	3 (18%)	0 (0%)
Social Anxiety Disorder	0 (0%)	3 (20%)
Specific Phobias	1 (6%)	2 (13%)
Generalised Anxiety Disorder	13 (81%)	14 (93%)

**Table 4 brainsci-12-01597-t004:** Comparison between the three groups in parent assessment scores separately for mothers and fathers.

Variable	MOTHERS				FATHERS			
	Group 1	Group 2	Group 3	*p*-Value	Group 1	Group 2	Group 3	*p*-Value
	ASD + AD	AD	ASD		ASD + AD	AD	ASD	
	N = 21	N = 31	N = 23		N = 21	N = 31	N = 23	
	Mean (SD)	Mean (SD)	Mean (SD)		Mean (SD)	Mean (SD)	Mean (SD)	
PSI-FS Total	85.57 (24.19)	62.58 (23.76)	70.61 (26.27)	0.002 *	65.95 (24.47)	63.06 (21.20)	51.52 (26.30)	0.101
SCL-90 GSI	59.34 (13.23)	46.93 (7.40)	54.13 (16.06)	0.002 *	60.29 (17.64)	47.55 (7.94)	46.35 (9.12)	0.001 *

PSI-SF: Parenting Stress Index-Short Form; SCL-90-R: Symptom Checklist 90-Revised. * *p* < 0.05.

## Data Availability

The datasets used and/or analysed during the current study are available from the corresponding author upon reasonable request. The raw data supporting the conclusions of this article will be made available by the authors, without undue reservation.

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
