# Peer review of "Anxiety in Autism Spectrum Disorder: Clinical Characteristics and the Role of the Family"

_brainsci, 2022, doi:10.3390/brainsci12121597_

Round 1
Reviewer 1 Report
The low language skills could also affect anxiety level. It would be important to include two informations in the manuscript in regard with the language and speech abilities of children:
1. It is unclear, if the language skills (Verbal IQ) had an impact or correlated with scores of anxiety. Please precise in the results section.
2. In accordance to verbal skills, please precise in the section Measures how many participants were assed with non-verabal intelligence scale.
Formatting issues:
In the Results section (lines 168-170) the extract comes probably from Author Guide. Please delete
Line 319, please correct formatting- the bolded extract 'suggested by Varela'.
Author Response
Manuscript ID: brainsci-2000273
Response to Reviewers' Comments:
We thank the reviewers for their scrutiny of the manuscript and insightful remarks, their very good feedback on our study was very encouraging. We hope to match their thoroughness and detail in our reply.
Please note that all changes made to the text altered the numbers of the lines you marked. Therefore, we have inserted our changes by referring to paragraph numbers.
Please note function that our replies are written in italics and that any changes to the text are marked up using the “Track Changes”
Response to Reviewer 1
-The low language skills could also affect anxiety level. It would be important to include two informations in the manuscript in regard with the language and speech abilities of children:
- It is unclear, if the language skills (Verbal IQ) had an impact or correlated with scores of anxiety. Please precise in the results section.
Response: Thanks for your suggestion. In our study, we did not perform statistical analyzes on the relationship between Verbal IQ and score of anxiety.
The IQ level was used only as an inclusion /exclusion criterion.
In our study, we focused on the clinical features of anxiety in ASD and we have tried to exclude those patients in whom anxiety could be related to the presence of a comorbidity with an intellectual disability.
We specify this better in the text (4.2 Strengths and limitations section):
“Considering that at least 50% of the ASD population has an IQ of less than 70, we are aware that this choice eliminates a substantial group of individuals with ASD. On the other hand, this choice allowed us to exclude clinical characteristics of anxiety related to other comorbidities, such as intellectual disability. Therefore, specific studies on IQ-impaired samples could be carried out in the future”
- In accordance to verbal skills, please precise in the section Measures how many participants were assed with non-verbal intelligence scale.
Response: Thanks for your request of clarification.
We have modified the text (2.2.1. Children and adolescents assessment) as follows:
“Finally, based on the patient’s collaboration and language development, cognitive functioning (IQ) was assessed by Wechsler Intelligence Scale for Children Fourth Edition (WISC-IV) [29] the Leiter-3 [30]or the Raven Matrix [31]. Specifically, in our sample, two participants were evaluated with the Leiter-3 [30], five with Raven Matrix [31] and sixty-eight with the WISC-IV [29].”
- Formatting issues:
- In the Results section (lines 168-170) the extract comes probably from Author Guide. Please delete
Response: Thanks for your suggestion. We have deleted the text “This section may be divided by subheadings. It should provide a concise and precise description of the experimental results, their interpretation, as well as the experimental conclusions that can be drawn”.
- Line 319, please correct formatting- the bolded extract 'suggested by Varela'.
Response: Thanks for your suggestion. We have corrected the text.

Reviewer 2 Report
Study Title: Anxiety in Autism Spectrum Disorder: Clinical characteristics and the role of the family
Overall
1. The author conducted ANOVA and t-tests/chi-square to investigate the between-group differences. I think besides the group comparisons, more information could be revealed if the authors conduct correlations between the Children’s and parents’ assessments. For example, correlations between the child’s ADOS scores and the MASC-2/CDI could tell us if children with greater ASD symptoms might have greater levels of anxiety and depression. Moreover, it’ll be interesting to see the associations between the child’s anxiety level and the parent’s stress index.
Introduction
1. At the end of the introduction section, instead of saying "Based on the literature described…” could the author provide the knowledge gaps that link to the aims of the study (Page 2, Line 77)?
2. Could the author provide more details about the aims and describe their hypothesis associated with the aims?
Materials and Methods
1. Please specify what is presented other than the participants’ age. For example, the author could say Mean age ± STD = 11.8 ± 2.3 (Page2, Line 83).
2. Could the author provide sex distribution of the including sample?
3. The author wrote, “All participants (N=75) were assessed with Social Communication Questionnaire (SCQ) [20], a screening instrument to confirm or exclude the presence of ASD.” (Page 3, lines 102 to 103). Based on my understanding, SCQ is a screening tool that screens the ASD symptoms without the capacity of confirming or excluding the diagnosis. Could the author modify the description to avoid confusion?
4. Was ADOS only done for children who showed ASD symptoms based on the SCQ score? If so, could the author specify the cut-off point of the SCQ score and who was evaluated using ADOS?
5. Could the author put the scores for K-SADS-PL DSM-5 in Table 1?
6. Could the author include the scores of the parent questionnaires (i.e., PSI-SF and SCL-90R) in Table 1?
7. This is just a minor suggestion for the format. I think the author could delete the subtitle “Measure” after “2.2.1 Children and adolescents assessment” and “2.2.2 Parent’s psychopathological distress clinical assessment”. And put “statistical analyses” up a level (i.e., 2.2.3).
Results
1. Please delete the first paragraph of the result section (Page 4, lines 168-170) as it is not associated with the current study.
2. The information described in Lines 172 to 180 is repetitive. Please decide where to keep the information, method, or the result session.
3. In multiple places, the author used the word “major” to describe the score. For example, in line 200, the author wrote “Group 1 reported major total score compared to Group 2 and Group 3”. I suggest using the word “higher” when comparing scores between groups.
Discussion
1. The second paragraph in the discussion session is very long. I recommend splitting it into several paragraphs.
2. In the first sentence, the author wrote “The first aim of our study was to deepen the clinical profile of children and adolescents with ASD and anxiety disorders by comparing three groups of children and adolescents (Group 1: ASD+AD, Group 2: AD, Group 3: ASD) matched by age, IQ and severity symptoms of ASD (Group 1: ASD+AD, Group 3: ASD).” Could the author specify what is being compared between the three groups?
3. Minor suggestions: i) I recommend adding a subtitle “4.1 Clinical implication and future directions” between Lines 330 and 331. ii) Please remove the bold font on line 319.
Author Response
Manuscript ID: brainsci-2000273
Response to Reviewers' Comments:
We thank the reviewers for their scrutiny of the manuscript and insightful remarks, their very good feedback on our study was very encouraging. We hope to match their thoroughness and detail in our reply.
Please note that all changes made to the text altered the numbers of the lines you marked. Therefore, we have inserted our changes by referring to paragraph numbers.
Please note function that our replies are written in italics and that any changes to the text are marked up using the “Track Changes”
Response to Reviewer 2
-This manuscript focused on a crucial topic since, based on available data, people with Autism Spectrum Disorders (ASD) are more prone to experiencing anxiety. As mentioned in the manuscript, probably half of all people with ASD experience high anxiety levels regularly. Therefore, the topic is worth considering from different perspectives and various forms.
- In the methodology section, line 83, it seems that the Standard Deviation has been deleted, and the present form of the central tendency reporting is unclear. A similar missing has been repeated in line 172 for all three groups.
Response: Thanks for your request of clarification. We have corrected the text (2.1 participants paragraph and 3. results section) as follows:
“(Mean age: 11.8 years, standard deviation (SD): 2.3)”
-The main question is the number of scales used for the study. Particularly those who have overlaps with each other. I think providing reasons for using the following scales to collect children's data:
- SCQ and ADOS-2 because of high standards of ADOS-2 application of SCQ (A 40 items scale) were not reasonable, and application of some more superficial screening scales might have been more logical. Also, the main question is why both screening and diagnosis scales were reported. While a "golden standard" scale for diagnosis was used?
Response: Thanks for your request of clarification.
We have modified the text (2.2.1 Children and adolescents assessment) as follows:
“In order to reach a diagnosis of ASD on the basis of criteria used in clinical practice (both directly administered evaluation and parent report) we choiced to include both SCQ and ADOS assessment.”
Indeed, as suggested by Reviewer 2, the ADOS is a a semi-structured assessment tool considered a “gold standard” for collecting standardized information about social communication skills, restricted interests and repetitive behaviors. It was administered directly to the patient and scored by licensed clinicians who have demonstrated clinical proficiency on the instrument. The SCQ is a parent-completed ASD screening tool of communication and social functioning derived from ADI-R, that is considered an other “gold-standard” instruments used for diagnosis of ASD, as referenced in the text.
- Schedule for Affective Disorders for evaluating Anxiety and Multidimensional Anxiety Scale for Children-Second Edition (MASC – 2). What was extra information that the application of the parental scale provided?
Response: Thanks for your suggestion.
We have modified the text (4. Discussion section) as follows:
“In line with current evidence that recommends on the use of multiple assessment modalities and informants to assess anxiety in children with ASD (Wood JJ et al. 2010), we proceded to include data from clinical interviews and rating scales gathered from multiple informants (e.g. child and parent).”
- Children's Global Assessment Scale (CGAS)
Response: Thanks for your suggestion. We have modified the text (4. Discussion section) as follows:
“To evaluate how much does anxiety symptoms interfere with daily functioning and if this impairment involve one or more contexts.”
- Children's Depression Inventory-2
Response: Thanks for your suggestion. We have modified the text (4.0 Discussion section) as follows:
“The use of parent and child versions of CDI-2 is widely applied to investigate depressive symptoms in children and adolescents with ASD”.
- Wechsler, Raven, and Leiter-3 all for the IQ evaluation, while each scale evaluates a different aspect of IQ. How could the finalresults be comparable? While Wechsler considers various verbal and non-verbal IQ factors, Leitre evaluates the non-verbal aspects of IQ and Rayven Matrixes logical thinking through visual stimulus. I am sure applying one scale might have yielded more reliable scores.
Response: Thanks for your suggestion. We have modified the text (2.2.1. Children and adolescents assessment) as follows:
“Finally, based on the patient’s collaboration and language development, cognitive functioning (IQ) was assessed by Wechsler Intelligence Scale for Children Fourth Edition (WISC-IV) [29] the Leiter-3 [30]or the Raven Matrix [31]”.
Also we have modified the text (4.2 Strengths and limitations section) as follows:
“Moreover, IQ was assessed by three different scales based on the patient’s collaboration and language development. Studies including homogeneous groups and therefore the use of a single scale of cognitive functioning, are desirable”.
- I also think it is crucial to provide some information regarding considering an IQ above 70 as an inclusion criterion for children with ASD. At the same time, all the scales on anxiety were through the parental report, and ASD scales were also applicable to individuals with ASD with different IQ levels. It is also necessary to mention that at least 50% of the ASD population has a lower than 70 IQ. What was the benefit of these criteria while you used IQ and ASD scales that were flexible enough to evaluate people at different IQ levels?
Response: Thanks for your suggestion. As suggested by reviewer 2 this limitation has now been better commented in the text (4.2 Strengths and limitations section) as follows:
“Considering that at least 50% of the ASD population has an IQ of less than 70, we are aware that this choice eliminates a substantial group of individuals with ASD. On the other hand, this choice allowed us to exclude clinical characteristics of anxiety related to other comorbidities, such as intellectual disability. Therefore, specific studies on IQ-impaired samples could be carried out in the future.”
- Since the study was done in Italy, were the Italian version of the scales available to be used?
Response: Thanks to request of clarification. We have specified this better in the text (2.2.1. Children and adolescents assessment paragraph and 2.2.2. Parents’ psychopathological distress clinical assessment paragraph):
“For all the measures used, the versions of the test validated in Italian were proposed to all the participants.”
- The study focuses on parents; hence no data has been presented regarding their demographic data. If no data regarding parents' gender, age, and socioeconomic background is collected, it is worth it to be mentioned in the limitations since each of these factors, similar to the child-mentioned characteristics, might have contributed to the results.
Response: Thanks for your suggestion. We have modified the text (4.2 Strengths and limitations section) as follows:
“Finally, no data has been collected regarding parent’s demographic features, such us gender, age and socioeconomic background. Therefore it may be appropriate to expand these details to further investigate the relationship between demographic features and parental psychopathological distress in this clinical population.”
- I am also keen to know the authors' justification regarding the reported SCQ results in confirming the statistically significant difference between the three groups. In contrast, ADOS-2 results might indicate the difference between groups 1 and 3.
Response: We specifically this better in the text (2.2.1. Children and adolescents assessment paragraph)
“All participants (N=75) were assessed with Social Communication Questionnaire (SCQ) [20], a screening instrument helps evaluate communication skills and social functioning. SCQ is a caregiver-report, derived from Autism Diagnostic Interview-Revised (ADI-R)[21] used to assess social communication impairment, the presence of repetitive and restrictive behaviors and screen ASD symptoms. The cut-off recommended by the SCQ manual [20] was ≥15. Of the 75 participants, 44 scored at or above this cut-off. For these, the presence of ASD was assessed by the Autism Diagnostic Observation Schedule-Second Edition (ADOS-2)[22]. It is a semi-structured assessment tool considered a “gold standard” for collecting standardized information about social communication skills, restricted interests and repetitive behaviors. ADOS-2 was administered and scored by licensed clinicians who have reached clinical reliability on the instrument. The calibrated severity score of each domain was also calculated [23,24].In order to reach a diagnosis of ASD on the basis of criteria used in clinical practice (both directly administered evaluation and parent report) we choiced to include both SCQ and ADOS assessment.”
According to this explanation, in table 1 you can see:
-Statistically significant difference between three groups. Specifically, Group 2 (participants with anxiety disorders without ASD) reported lower scores compared to Group 1 and Group 3.
- Not significant differences were found between the Group 1 and Group 2 in ADOS-2 total score and in ADOS-2 score of dimensions.
- I want to stress that a confirmed diagnosis of ASD was mentioned as one of your inclusion criteria, while group two in your study had no ASD diagnosis! This criterion needs to be amended.
Thanks for your request of clarification. We have modified the text (2.1 participants paragraph) as follows:
“We included participants with only ASD diagnosis, participants with only AD diagnosis and participants with diagnosis of both. Our diagnosis follow on the DSM-5 criteria (APA, 2013).For all participant the inclusion criteria was an Intellectual Quotient (IQ) higher than 70.”
- In sum, I think that the paper, based on its title supposed to focus on family characteristics and anxiety in children with ASD equally, but the present format of the manuscript is skewed towards children's diagnosis, comorbid conditions, and the level of collected data and analysis on children's information is very rich and extensive that the parental section and the impacts of anxiety on their PPD are very minimally considered or covered. I even think that based on the collected data, two separate papers might possibly be developed (of course, this is the authors' decision). Still, addressing the abovementioned issues may help the authors present their valuable data in a more equally focused approach.
Response: Thanks for your suggestion. As suggested by reviewer 2, we have modified the text (4.2 Strengths and limitations section) as follows:
“Therefore it may be appropriate to expand these parents’ clinical features, such as knowing much more about the impact of anxiety symptoms on their PPD. In sum, we hope that future studies’ goal will be investigating deeper the role of psychological profile of parents on psychiatric comorbidities in children and adolescents with ASD.”

Reviewer 3 Report
This manuscript focused on a crucial topic since, based on available data, people with Autism Spectrum Disorders (ASD) are more prone to experiencing anxiety. As mentioned in the manuscript, probably half of all people with ASD experience high anxiety levels regularly. Therefore, the topic is worth considering from different perspectives and various forms.
In the methodology section, line 83, it seems that the Standard Deviation has been deleted, and the present form of the central tendency reporting is unclear. A similar missing has been repeated in line 172 for all three groups.
The main question is the number of scales used for the study. Particularly those who have overlaps with each other. I think providing reasons for using the following scales to collect children's data:
· SCQ and ADOS-2 because of high standards of ADOS-2 application of SCQ (A 40 items scale) were not reasonable, and application of some more superficial screening scales might have been more logical. Also, the main question is why both screening and diagnosis scales were reported. While a "golden standard" scale for diagnosis was used?
· Schedule for Affective Disorders for evaluating Anxiety and Multidimensional Anxiety Scale for Children-Second Edition (MASC – 2). What was extra information that the application of the parental scale provided?
· Children's Global Assessment Scale (CGAS)
· Children's Depression Inventory-2
· Wechsler, Rayven, and Leiter-3 all for the IQ evaluation, while each scale evaluates a different aspect of IQ. How could the final results be comparable? While Wechsler considers various verbal and non-verbal IQ factors, Leitre evaluates the non-verbal aspects of IQ and Rayven Matrixes logical thinking through visual stimulus. I am sure applying one scale might have yielded more reliable scores.
· I also think it is crucial to provide some information regarding considering an IQ above 70 as an inclusion criterion for children with ASD. At the same time, all the scales on anxiety were through the parental report, and ASD scales were also applicable to individuals with ASD with different IQ levels. It is also necessary to mention that at least 50% of the ASD population has a lower than 70 IQ. What was the benefit of these criteria while you used IQ and ASD scales that were flexible enough to evaluate people at different IQ levels?
· Technically, individuals with ASD with an IQ above 70 are considered a high-functioning group and generally a vocal group but have other psychological comorbidities. Such as Repetitiveness. HFA is partly characterized by an obsession with a particular subject or activity, emotional sensitivity, social problems, language peculiarities, Sensory difficulties, and Little or no attention to caregivers.
· Since the study was done in Italy, were the Italian version of the scales available to be used?
· The study focuses on parents; hence no data has been presented regarding their demographic data. If no data regarding parents' gender, age, and socioeconomic background is collected, it is worth it to be mentioned in the limitations since each of these factors, similar to the child-mentioned characteristics, might have contributed to the results.
· I am also keen to know the authors' justification regarding the reported SCQ results in confirming the statistically significant difference between the three groups. In contrast, ADOS-2 results might indicate the difference between groups 1 and 3.
· I want to stress that a confirmed diagnosis of ASD was mentioned as one of your inclusion criteria, while group two in your study had no ASD diagnosis! This criterion needs to be amended.
In sum, I think that the paper, based on its title supposed to focus on family characteristics and anxiety in children with ASD equally, but the present format of the manuscript is skewed towards children's diagnosis, comorbid conditions, and the level of collected data and analysis on children's information is very rich and extensive that the parental section and the impacts of anxiety on their PPD are very minimally considered or covered. I even think that based on the collected data, two separate papers might possibly be developed (of course, this is the authors' decision). Still, addressing the abovementioned issues may help the authors present their valuable data in a more equally focused approach.
Author Response
Manuscript ID: brainsci-2000273
Response to Reviewers' Comments:
We thank the reviewers for their scrutiny of the manuscript and insightful remarks, their very good feedback on our study was very encouraging. We hope to match their thoroughness and detail in our reply.
Please note that all changes made to the text altered the numbers of the lines you marked. Therefore, we have inserted our changes by referring to paragraph numbers.
Please note function that our replies are written in italics and that any changes to the text are marked up using the “Track Changes”
Response to Reviewer 3
- The author conducted ANOVA and t-tests/chi-square to investigate the between-group differences. I think besides the group comparisons, more information could be revealed if the authors conduct correlations between the Children’s and parents’ assessments. For example, correlations between the child’s ADOS scores and the MASC-2/CDI could tell us if children with greater ASD symptoms might have greater levels of anxiety and depression. Moreover, it’ll be interesting to see the associations between the child’s anxiety level and the parent’s stress index.
Response: Thank you. We find your suggestion very interesting. However, we did not carry out correlational analyzes given the small clinical sample. We plan to do a correlational study when we have larger samples.
- At the end of the introduction section, instead of saying "Based on the literature described…” could the author provide the knowledge gaps that link to the aims of the study (Page 2, Line 77)? Could the author provide more details about the aims and describe their hypothesis associated with the aims?Could the author provide more details about the aims and describe their hypothesis associated with the aims?
Response: Thanks for your suggestion. We have modified the text (1. Introduction) as follows:
“Also, the role of Parental Psychological Distress (PPD) in families of children and adolescents with ASD and AD should not be underestimated. PPD is defined as family members’ distress, resulting in high levels of family conflict, and increased expressed emotions, defined as criticism, hostility, and emotional over-involvement [10,11]. Parents of children with ASD have higher parental stress levels and lower quality of life than parents of normotypical children [12-15], similarly to parents with disabilities’ children (e.g., Down syndrome, cerebral palsy, and intellectual disability) [16-18]. Yorke et al.[19], in a systematic review, examined the relationships between the children’s emotional and behavioral problems and PPD in a group of ASD children, finding a moderate association between parenting stress, parental mental health problems, and ASD child’s emotional and behavioral problems. Taken together, these studies have focused mainly on the prevalence of psychiatric comorbidities in children and adolescents with ASD neglecting their clinical characteristics and significance and the role of PPD. Concerning anxiety disorders in ASD, to identify specific treatments, it is crucial to understand which anxiety disorders are most frequently observed and clarify their peculiar clinical characteristics in children and adolescents with ASD. Also, it is important to explore the presence or not of impairment in the global functioning (e.g. social, scholastic, and family contexts) associated with anxiety disorder and the role of parental psychological profile. Based on this, the main aim of this study was to deepen the clinical profile of children and adolescence with the co-occurrence of ASD+AD focusing on the characteristics of anxiety disorders in this clinical population, the functional impairment associated, and the possible presence or not of high levels of PPD. We propose to test the hypothesis that anxiety disorders in ASD can be considered a separate and distinct construct from the main symptoms of ASD and therefore worthy of clinical attention through interventions aimed at children or adolescents but also at their parents.”
- Please specify what is presented other than the participants’ age. For example, the author could say Mean age ± STD = 11.8 ± 2.3 (Page2, Line 83).
Response: Thanks for your request of clarification. We have corrected the text (2.1 participants paragraph and 3. results section) as follows:
(Mean age: 11.8 years, standard deviation (SD): 2.3)
- Could the author provide sex distribution of the including sample?
Response: Thanks of your suggestion. We have modified the text (2.1 participants paragraph and 3.1 sample characteristics paragraph) as follows:
“75 children and adolescents (mean age: 11.8, standard deviation (SD): 2.3 years; Range age: 8-16 years; Males: 64; Females: 11)”
In the paragraph 3.1 Sample of characteristics you can see sex distribution of three groups.
- The author wrote, “All participants (N=75) were assessed with Social Communication Questionnaire (SCQ) [20], a screening instrument to confirm or exclude the presence of ASD.” (Page 3, lines 102 to 103). Based on my understanding, SCQ is a screening tool that screens the ASD symptoms without the capacity of confirming or excluding the diagnosis. Could the author modify the description to avoid confusion? Was ADOS only done for children who showed ASD symptoms based on the SCQ score? If so, could the author specify the cut-off point of the SCQ score and who was evaluated using ADOS?
Response: Thanks for your suggestion and your request of clarification. We have modified the text (2.2.1 Children and adolescents assessment paragraph) as follows:
“All participants (N=75) were assessed with Social Communication Questionnaire (SCQ) [20], a screening instrument helps evaluate communication skills and social functioning. SCQ is a caregiver-report, derived from Autism Diagnostic Interview-Revised (ADI-R)[21] used to assess social communication impairment, the presence of repetitive and restrictive behaviors and screen ASD symptoms. The cut-off recommended by the SCQ manual [20] was ≥15. Of the 75 participants, 44 scored at or above this cut-off. For these, the presence of ASD was assessed by the Autism Diagnostic Observation Schedule-Second Edition (ADOS-2)[22]. It is a semi-structured assessment tool considered a “gold standard” for collecting standardized information about social communication skills, restricted interests and repetitive behaviors. ADOS-2 was administered and scored by licensed clinicians who have reached clinical reliability on the instrument. The calibrated severity score of each domain was also calculated [23,24].In order to reach a diagnosis of ASD on the basis of criteria used in clinical practice (both directly administered evaluation and parent report) we choiced to include both SCQ and ADOS assessment.”
- Could the author put the scores for K-SADS-PL DSM-5 in Table 1?
Response: Thanks for your request of clarification. K-SADS PL dsm-5 is a semi-structured interview based on DSM-5 criteria. Due to large number of symptoms of psychiatric disorders explored (in line with DSM-5 criteria), it is not possible to include the score for all participants. In addiction, the score on the single item/criteria is not indicative of a diagnosis of a given psychiatric disorder. This interview is based on clinical judgment and on the rule that all DSM-5 criteria for a given psychiatric disorder are met. However, we have inserted the tables with the frequencies and percentage frequencies of the diagnoses of anxiety disorder made in the three groups with the K- SADS PL DSM-5.
In table 2 and 3 we summarized the results included in the paragraph Comparison between Group 1 and Group 2 in Anxiety Clinical Profile.
- Could the author include the scores of the parent questionnaires (i.e., PSI-SF and SCL-90R) in Table 1?
Response: Thanks for your request of clarification. Due to the large number of items in both questionnaires, it is not possible to include the score of all participants in both questionnaires. In Table 4, we have included the average of the total scores on both questionnaires for parents of the three groups.
- This is just a minor suggestion for the format. I think the author could delete the subtitle “Measure” after “2.2.1 Children and adolescents assessment” and “2.2.2 Parent’s psychopathological distress clinical assessment”. And put “statistical analyses” up a level (i.e., 2.2.3).
Response: Thanks. Based on your suggestion for the format, we have modified the text.
- Please delete the first paragraph of the result section (Page 4, lines 168-170) as it is not associated with the current study.
Response: Thanks. Based on your suggestion, we have deleted the text.
- The information described in Lines 172 to 180 is repetitive. Please decide where to keep the information, method, or the result session.
Response: Thanks for your suggestion. We have included this information only in results section (3.1 sample characteristics paragraph).
- In multiple places, the author used the word “major” to describe the score. For example, in line 200, the author wrote “Group 1 reported major total score compared to Group 2 and Group 3”. I suggest using the word “higher” when comparing scores between groups.
Response: Thanks. Based on your suggestion, we have modified the text.
- The second paragraph in the discussion session is very long. I recommend splitting it into several paragraphs.
Response: Thanks. Based on your suggestion, we have tried to modify the text in order to make it more fluid.
- In the first sentence, the author wrote “The first aim of our study was to deepen the clinical profile of children and adolescents with ASD and anxiety disorders by comparing three groups of children and adolescents (Group 1: ASD+AD, Group 2: AD, Group 3: ASD) matched by age, IQ and severity symptoms of ASD (Group 1: ASD+AD, Group 3: ASD).” Could the author specify what is being compared between the three groups?
Response: Thanks for your suggestion. We have simplified the text (4. Discussion) as follows:
“The first aim of our study was to deepen the clinical profile of children and adolescents with ASD and anxiety disorders focusing on the characteristics of anxiety in this clinical population, the functional impairment associated, and the possible presence or not of high levels of PPD. Based on this goal, we compared three groups of children and adolescents (Group 1: ASD+AD, Group 2: AD, Group 3: ASD) matched by age, IQ and severity symptoms of ASD.”
- Minor suggestions: i) I recommend adding a subtitle “4.1 Clinical implication and future directions” between Lines 330 and 331. ii) Please remove the bold font on line 319.
Response: Thanks for your suggestion. We have modified and corrected the text

Round 2
Reviewer 3 Report
This is an improved version of the previously submitted manuscript. The authors have done their best to address the comments and suggestions of all three reviewers.